# Personal and Job-Related Factors Influencing the Work Engagement of Hospital Nurses: A Cross-Sectional Study from Saudi Arabia

**DOI:** 10.3390/healthcare11040572

**Published:** 2023-02-15

**Authors:** Hanan Alkorashy, Manar Alanazi

**Affiliations:** 1Nursing Administration & Education Department, College of Nursing, King Saud University, Riyadh 12372, Saudi Arabia; 2Nursing Administration Department, Faculty of Nursing, Alexandria University, Alexandria 21544, Egypt; 3Executive Nursing Affairs, Prince Sultan Military Medical City, Riyadh 12233, Saudi Arabia

**Keywords:** work engagement, hospital nursing, Saudi Arabia, Utrecht Work Engagement Scale

## Abstract

This study explored the levels of work engagement and identified whether personal and job-related factors influenced the work engagement dimensions of vigor, dedication, and absorption of nurses working in a Saudi hospital. A descriptive, cross-sectional correlational survey of nurses in inpatient wards (general medical, surgical, and specialized wards) and critical care units in a tertiary hospital in Saudi Arabia, using The Utrecht Work Engagement Scale. Using a self-report questionnaire, 426 staff nurses and 34 first-line nurse managers were surveyed. Data collected consisted of selected personal and professional factors, including gender, age, education, current work setting, years of experience, nationality, and participation in committees, and/or work teams alongside the 17-item version of the UWES. The study participants showed high levels of work engagement. Age, years of experience, and participation in committees were significantly associated with work engagement. Nurses who were older, possessed more experience, and participated in committees showed higher levels of engagement. Healthcare organizations and their leaders, policymakers, and strategic planners should create a conducive work environment that supports the work engagement of nurses by considering the influencing antecedents. The nursing profession, patients’ safety issues, and vital economic problems are the fundamental issues facilitated by the creation of practice environments that entirely engage nurses in their work.

## 1. Introduction

The world is currently facing a shortage of nurses, and consequently, an inability to achieve and sustain universal health coverage [1] and Saudi Arabia is no exception [2,3]. Thus, healthcare organizations make extraordinary efforts to retain their distinguished taskforces, especially nurses, and inspire them to render quality and safe care to diverse populations of patients [4]. The effective utilization of the nursing taskforce, among other healthcare professionals, can bring sustainable and competitive advantages to the organization. An important way in which such utilization of human resources could be delivered is by engaging them in their work [4,5]. In this context, an integrative review pointed out that healthcare bodies should offer working conditions that nurture work engagement to ensure that nurses remain engaged in their work [6]. Moreover, those factors that contribute to a high level of engagement should be considered as a way of improving the overall performance of healthcare providers, especially nurses [6], the overall healthcare outcomes, and patient safety [7,8].

Engagement was first conceptualized and defined by Kahn as “The ‘harnessing of organizational members’ selves to their work roles; in engagement, people employ and express themselves physically, cognitively and emotionally during role performances” [9]. Harter et al. further defined employee engagement as “the individual’s involvement and satisfaction as well as enthusiasm for work” [10]. Thus, employee engagement is the level of commitment and involvement an employee has towards his or her organization and its values.

Work engagement, as conceptualized by the Utrecht Work Engagement Scale (UWES), is a work-related positive state of mind. Work engagement is composed of three dimensions: vigor, dedication, and absorption. The first dimension, “vigor” is defined as “high levels of energy and mental resilience while working, the willingness to invest effort in one’s work, and persistence even in the face of difficulties” [11]. Vigor implies to when an individual has high levels of energy and mental resilience, total investment in the actual work, and is highly persistent even during difficulties [12]. The second dimension, “dedication” is defined as “a sense of significance, enthusiasm, inspiration, pride, and challenge” [11]. Dedication is when an individual derives a sense of significance from work, feeling enthusiastic, proud, and inspired about the given job, as well as challenged by the job [12]. The final dimension, “absorption” is defined as “being fully concentrated and deeply engrossed in one’s work, whereby time passes quickly, and one has difficulties with detaching oneself from work” [9]. Furthermore, the job experience is pleasurable for individuals. They enjoy their job and a high workload does not deter them, unlike others [13].

Prior studies have shown that work engagement can improve the quality of nursing services by motivating nurses to maximize their competence and expertise [14,15]. Nurses that experienced high levels of job engagement showed an increase in caring behavior, job satisfaction, and employee productivity [16,17]. Nurse engagement levels have also been associated with low burnout and decreased turnover intentions [18].

Staff shortages and high nurse turnover are major concerns in the current nursing industry across the country and the globe [3].

As in many other countries, nurse turnover rates affect the ability of hospitals in Saudi Arabia to effectively address patient needs and deliver quality and safe care [19]. Staff shortages due to high nurse turnover rates result in excessive and unhealthy workloads and stress for the remaining staff. Additionally, nurses may be required to perform disparate roles, resulting in low productivity [2,3].

Concomitantly, additional consideration is related to the large number of non-Saudi nurses working in Saudi hospitals [3], who are familiar with the Saudi culture [19], however, they experience a less favorable attitude from some Saudi citizens because of the perceived cultural differences [20]. Such prejudices intensify stress among these nurses, hampering their work engagement [3].

A significant amount of research has been conducted on work engagement, among which, studies have reported that work engagement can have a positive impact on various aspects of individuals [15,21], and organizations [22,23].

In Spain, a cross-sectional study was conducted to assess the level of work engagement of Spanish nurses during the COVID-19 pandemic, it was concluded that the Spanish nurses presented high levels of work engagement in all dimensions in general, although the lowest mean scores were found in the vigor dimension among male nurses and nurses working in hospital and critical units [24]. When engaged in their work, employees enter into an interactive mode comprising challenges, inspirations, and pride, which contributes to their job satisfaction. Work engagement is the fifth line of inquiry in the American Academy of Nursing’s magnet study of hospitals, and an emerging research focus area in positive psychology. Nurses, in their sample, showed not only high work engagement overall, but also in all specific dimensions, namely: vigor, absorption, and dedication [24]. Another qualitative study conducted in Spain aimed to establish the factors that help nursing professionals to engage with their work environment. The study concluded that there are different proposals that act as transformative elements favoring the appearance of engagement among nurses. These include training nurses on effective communication, emotional management, and social skills, enhancing management efforts to foster a feeling of belonging and justice at the workplace, and improving the recognition and reward of nursing professionals, both institutionally, socially, and in the family [25].

A cross-sectional study of 425 Japanese psychiatric nurses conducted in Japan aimed to identify the factors influencing work engagement. The results revealed that the self-motivation dimension of emotional intelligence played a particularly important role in enhancing work engagement among psychiatric nurses. Moreover, nurses with a positive perception of reward, supervisor support, and nurse–physician collaboration tended to possess a higher work engagement [26].

A questionnaire-based study conducted in China explored the work engagement levels of dental nurses and identified key factors that potentially influence work engagement. The key findings in the study showed that nurses had an acceptable level of work engagement in terms of vigor, dedication, and absorption; increased job stress resulted in lower work engagement and nurses who had higher levels of perceived social support, psychological flexibility, and subjective wellbeing also had a higher work engagement [27].

In Saudi Arabia, researchers considered work engagement in their studies either as a mediating factor or as an outcome [28,29,30,31]. A cross-sectional study assessed the level of work engagement among nurses in the Ministry of Health hospitals in Najran, Saudi Arabia, and the factors that influence their work engagement. This study found that 49% of the participants were moderately engaged in their work and that personal attributes, organizational factors, and leadership factors were all positively and significantly correlated with work engagement [32].

Engaged employees who approach their work with great effort, dedication, and concentration are more responsible, productive, and inventive, they are also more willing to offer voluntary services that are not part of their official duties [33]. Work-family enrichment enhances the work engagement levels of nursing personnel [34]. As mentioned earlier, support from supervisors also plays a significant role in increasing work engagement. However, it is not always workplace resources or practices that enhance nurses’ productivity, while their family life also plays a crucial role [35].

Usually, nurses are enthusiastically active participants at work, which enhances their performance and promotes the interests of their organization. Such active participation occurs when nurses consider the significance of their work, have control over their practice, have good working relationships, feel fairly treated, and are rewarded for their contributions to their organizational achievements [33]. In contrast, nurses who enjoy less social recognition and power in their work exhibit lower levels of work engagement [25]. Moreover, employer and workplace-related factors enhance the performance of nurses [35]. Promoting nursing teamwork as a job resource may effectively improve the nurses’ work engagement [36]. Therefore, this study’s findings provide insight into the personal and job-related factors and measures that can be taken by healthcare organizations, policymakers, and nursing leaders to improve nurses’ level of engagement in their work, and consequently, impact their organizational performance positively, and improve the quality, sustainability, and safety of rendered health services. We believe that our study makes a significant contribution to the literature because it investigates a highly relevant issue, namely nurses’ work engagement, against the backdrop of an ongoing global shortage of nurses and constant high demand for quality healthcare. Further, its findings enhance existing knowledge about personal and professional factors that impact not only work engagement in general, but also in terms of the three specific dimensions of vigor, dedication, and absorption. We substantiated the findings of several relevant previous studies. Consequently, its findings have future bearings on healthcare planning and management.

Thus, this study aimed to explore the level of work engagement and whether sociodemographic and job-related factors influence the work engagement dimensions of vigor, dedication, and absorption of nurses working in Saudi hospitals. The following research questions were formulated: (1) What is the level of work engagement among nurses in the selected setting? (2) Is work engagement influenced by the personal and/or professional characteristics of nurses?

## 2. Materials and Methods

### 2.1. Design

A cross-sectional, correlational questionnaire survey was conducted among nurses working in general in-patient wards, critical care units, and specialized wards (such as hematology, nephrology, specialized surgeries, etc.) in a large tertiary hospital located in Riyadh city—the capital of Saudi Arabia—which explored their levels of the work engagement and identified the key factors that potentially influenced their level of engagement.

### 2.2. Participants

A quota sample was taken from among the staff nurses at the target hospital (*n* = 1840). The minimum sample size was calculated using the G*Power 3.0 software (BSC702, Germany). For an effect size of 0.3 (medium), a significance level (α) of 0.05, and a test power (1 − β) of 0.95, the minimum number of participants required was 488. Given the possibility of invalid questionnaires, the sample size was increased by approximately 10% to account for non-responses and potential dropouts, bringing the final target sample size to 550. Regardless of their department and shift, all staff nurses working in an in-patient general or specialized ward, or in a critical care unit, currently providing direct patient care, able to read and write in English, employed at the target hospital for at least one year, and willing to participate in this study were included.

Most participants were female (97.4%), aged between 20 and 40 years (89.8%), and with a bachelor’s degree (89.8%). Furthermore, approximately half of them were working in general wards (58.7%), had 5–9 years of experience (53.9%), and had participated in clinical departmental or hospital-wide committees (60%). Table 1 shows the descriptive statistics for the participants’ characteristics.

### 2.3. Data Collection

After obtaining the official permission from the developers of the Utrecht Work Engagement Scale (UWES) to use their questionnaire, and the approvals from the data collection settings, the validity of the tool was established by submitting the questionnaire to a jury panel of five academicians and professionals in the same field of nursing management. Moreover, to ensure the reliability and consistency of the questionnaire, estimate the time required to complete the questionnaire, and ensure the clarity of the statements, a pilot study was conducted with 50 nurses (approximately 10% of the sample size) from three general wards, which were not involved in the main data collection settings. No concerns pertaining to the clarity and consistency of the questionnaire’s content were reported by the participants. The researchers met with the target hospital’s nursing administrator and the head nurses of the units and wards to explain the purpose of the study and obtain their assistance. 

The questionnaire and consent forms were distributed to all potential participants. Participation was voluntary. Participants were informed about their right to withdraw from the study at any stage and their confidentiality was assured. They were also told that their information would solely be used for the purposes of this study. All participants provided written informed consent. The average time required to complete the questionnaire was 12–15 min. Data collection took place over three months (March to May 2019).

### 2.4. Instrument

A self-report questionnaire was used to collect data, which consisted of two parts: (1) Selected personal and professional factors, including gender, age, education, current work setting, years of experience, nationality, and participation in committees and/or work teams; (2) the 17-item version of the UWES [37], wherein all items were measured using a 7-point Likert scale ranging from 0 (“never”) to 6 (“always”). The total score for each nurse on the UWES was calculated to determine overall reported work engagement, with higher scores indicating higher engagement. Cutoff values were statistically specified to give a clear meaning for the calculated scores for high (4–6), moderate (2 < x < 4), and low (0 < x < 2) reported engagements. Work engagement is assessed based on three dimensions: physical (vigor; VI), emotional (dedication; DE), and cognitive (absorption; AB). The VI dimension refers to the levels of energy, resilience, stamina, persistence, and willingness to invest effort. The DE dimension refers to finding significance and pride in one’s work, enthusiasm for one’s work, and feeling inspired and challenged by one’s work. Finally, the AB dimension refers to being completely and happily immersed in one’s work and having difficulty leaving, because time seems to pass quickly, and other commitments and responsibilities are forgotten [37]. The UWES has been used across multiple sectors of work in numerous studies worldwide [37,38]. Cronbach’s alpha for the composite scale was calculated in previous studies as 0.90, with the subscale alphas all exceeding 0.80 [11,37,38,39,40]. In the current study, Cronbach’s alpha for the UWES was 0.95, and for the subscales vigor, deduction and absorption were 0.88, 0.93, and 0.87, respectively.

### 2.5. Statistical Analysis

The demographic data were expressed as frequencies and percentages. Descriptive statistics, including frequency, percentage, mean, and standard deviation (SD) were used to explore the profile of the participants and determine each nurse’s level of work engagement. Perceived work engagement was a dependent variable, whereas personal and professional factors were independent variables. First, the univariate analysis included an independent t-test, then, a multivariate analysis of variance (MANOVA) was also conducted to examine the association between work engagement and the participant’s personal and professional characteristics. A *p*-value of ≤0.05 was considered statistically significant.

## 3. Results

Of the 550 questionnaires distributed, 524 bedside nurses responded to the survey, yielding a response rate of 95.3%. Of these, 64 were excluded because of either incomplete or invalid responses. Thus, the final number of questionnaires included in the analysis was 460.

Table 2 illustrates the nurses’ levels of work engagement. The participants showed a high level of work engagement, both overall (M = 5.47, SD = 0.908) and across all three dimensions. The highest mean level of work engagement was found for the DE dimension (M = 5.68, SD = 1.027), while the lowest mean (M = 5.27, SD = 0.968) was recorded for the AB dimension.

Table 3 displays the association between the nurses’ level of work engagement and personal characteristics. There was no statistically significant association between the level of work engagement and gender (t = −0.222, *p* = 0.824), indicating similar levels of work engagement among male and female nurses.

The only personal factor that had a statistically significant association with the level of work engagement was age (F = 11.128, *p* < 0.001), where the nurses’ level of work engagement increased as they became older. This positive association was apparent in all three dimensions of work engagement.

The educational background had no statistically significant correlation with the level of work engagement (F = 0.320, *p* = 0.726). Participants with a bachelor’s degree showed the highest level of overall work engagement in each of its dimensions (M = 5.58, SD = 1.04) compared with participants with diplomas (M = 5.47, SD = 0.90), and those with degrees higher than a bachelor’s degree (M = 5.37, SD = 0.93). 

Additionally, there were no statistically significant differences between Saudi (M = 5.31, SD = 1.16) and non-Saudi nurses (M = 5.49, SD = 0.87) regarding the overall level of work engagement or its dimensions, although the non-Saudi nurses showed a higher level of engagement (M = 5.49, SD = 0.87) than the Saudi nurses.

Table 4 shows the associations between the level of work engagement and professional factors. The findings revealed a statistically significant positive association between work engagement and years of experience (F = 14.081, *p* = 0.001); nurses’ work engagement increased with the length of stay in their work setting.

However, no statistically significant associations were found between the level of work engagement and work setting/specialty (F = 1.888, *p* = 0.153). Thus, although nurses in general wards (M = 5.54, SD = 0.90) recorded the highest level of work engagement compared with those in intensive care units (M = 5.36, SD = 0.82) and specialized units (M = 5.39, SD = 0.96), this difference was not statistically significant with all three groups showing similar levels of work engagement.

Finally, a statistically significant association was found between the level of work engagement and participation in committees (t = 4.102, *p* = 0.001). Nurses who participated in committees (M = 5.61, SD = 0.85) showed a significantly higher level of work engagement than those who did not participate in any committees (M = 5.27, SD = 0.95).

## 4. Discussion

This study examined nurses’ work engagement and highlighted specific personal and professional factors that impact their level of work engagement. The participating nurses exhibited high levels of overall work engagement and obtained high scores on each of its dimensions. These findings are consistent with many previous studies, including research conducted in Egypt [41], Spain [24,42], and the United States [43], which found high to very high levels of work engagement among nurses. The workplace environment, availability of motivators and incentives, and ability to make decisions (autonomy) were identified as significant predictors of a high level of work engagement among nurses [42,43]. Such a high level of work engagement can be explained by the likelihood that nurses who are committed to their roles have high energy and become absorbed in their work, which increases their self-realization and perceived meaningfulness of their work [24,42]. In this context, literature emphasized the magnitude of association between the worker’s high involvement in management and their sense of wellbeing [44,45,46]. Those organizations enrich employees’ working lives by offering them greater job autonomy, more mental stimulation, team-based social interaction, and an upraised sense of achievement, which may improve employees’ wellbeing [44]. However, the current findings differ from those of Alfifi et al., who conducted a similar study in two government hospitals in Saudi Arabia [32]. Here, they observed an average level of work engagement in 49% of their participants, low engagement in 20%, high engagement in 20%, very low engagement in 4%, and very high work engagement in just 5%. Meanwhile, Lourenção, who used the UWES to examine participants in nursing-focused residency and professional development programs, found a moderate level of VI and AB alongside a high level of DE and overall engagement [47].

The current study also found a statistically significant positive association between the level of work engagement and years of experience. The nurses’ work engagement increased with their work experience. This is in line with the findings of Saiga and Yoshioka, who found that work engagement was significantly associated with age, years of nursing experience, marital status, children, job position, and workplace [48]. Moreover, the present findings are also consistent with findings by Remegio, et al., who found that nurse leaders with over 20 years of experience had a higher level of work engagement than those with 5 years of experience or less [49]. This could be because more years of work experience provide nurses with more time to explore lesser-known aspects of their work and master their roles, enhancing knowledge that helps them become fully engaged in their work. Furthermore, nurses with more years of experience are likely to have attended many training programs, seminars, and workshops. Such programs contribute to their work engagement by equipping them with various techniques that can be applied to enhance their competence, and thus, elevate their performance. The improvement in performance also impacts their level of work engagement [50].

According to Remegio et al. [49], experienced nurse leaders with doctorates have higher levels of compassion, satisfaction, engagement, and lower levels of burnout and secondary traumatic stress. In the present study, participants with a bachelor’s degree showed the highest level of work engagement in all three dimensions (compared with participants with a diploma or a degree higher than a bachelor’s degree). Education and training play a vital role in helping nurses understand the nature of the services they provide. This may explain why nurses with a bachelor’s degree are more engaged in their work than those with lower-level certificates. Thus, a university education helps nursing students appreciate the significance of the nursing profession. By contrast, lower levels of work engagement shown by nurses with a higher level of education than a bachelor’s degree may be explained by the fact that nurse managers desire staff who can carry out practical work, rather than those deploying advanced skills in critical thinking and gathering knowledge to manage their patients. Kjellaas, et.al. [51] explored the reasons for disengagement among nurses with a master’s degree. One identified reason was that these nurses did not feel that their higher qualification was valued in their department of work, leading them to direct their efforts to teaching and mentoring staff, conducting scientific research, and looking for further development opportunities [51].

The current study also found a statistically significant association between the level of work engagement and age, where the nurses’ work engagement increased as they become older. This positive association was evident in all dimensions of work engagement. A similar result was reported in two previous studies [31,52] which revealed that the level of work engagement is strongly predicted by the nurse’s age, tenure, and educational level. Tomietto et al. [53] highlighted that the motivational factors for enhancing work ability differed between nurses aged <45 years and those aged ≥45 years. For nurses aged 45 or more, dedication was an important factor that motivated them to perform meaningful work and enhanced their work ability. For nurses younger than 45 years, vigor was an important motivational factor. This indicates that persistence in tasks at work and perceiving work as an activity to which one should devote time and effort are essential factors, motivating younger nurses to enhance their work ability. According to Huber and Schubert [54], younger nurses (Generation Y) are less loyal to their institutions than older nurses (Generation X and the baby boomers). However, Mhatre and Conger [55] clarified that what is sometimes described as disloyalty in this generational group may be their unwillingness to be loyal to an institution when they perceive it to be at the expense of their career growth and realization of professional goals. Huber and Schubert justified this conclusion by indicating that younger nurses desire a balance between their private life and professional involvement, better career opportunities and prospects, and a more comfortable work climate [54]. The current results can be explained from the perspective that with age, nurses understand the different aspects of the work environment better and appreciate the services they provide to physicians and patients alike, thus, helping them become fully engaged in their work. That is, as nurses age, they become more mature and aware of their duties and responsibilities, and consequently, become fully engaged in their clinical environments.

This study found no statistically significant association between the level of work engagement and gender, indicating that male and female nurses have a similar level of work engagement. This is consistent with the study by Saiga and Yoshioka, who also found no significant differences between male and female workers in terms of work engagement [48]. A possible reason for this is that both male and female nurses understand the importance of being fully engaged to improve patient care and work environment, and the gender of the healthcare provider does not influence their ability to be involved in all patient-related tasks.

### Limitations

This study has certain limitations. The first is related to the design of the study. The use of non-probability quota sampling with convenience samples being drawn from each stratum may have led to a bias concerning the representativeness of the sample. Second, because of the use of self-administered questionnaires, the accuracy of the data is subject to the honesty of the participants in their responses. Third, the time of sampling and the variations in working conditions across the sample (as the nurses were from general, critical care, and specialized wards) may have influenced the responses. Fourth, a deeper look at the work engagement (as a dependent variable) and the personal and professional factors (as independent variables) by providing further analysis may specify the factors that are mostly associated with the nurses’ work engagement. Finally, other variables related to work engagement, such as autonomy, control over nursing practice, turnover intention, and/or availability of organizational support, were not assessed in this study. Further research is needed to study current variables more in-depth and identify whether a causative relationship exists among any of the variables and the relationship between work engagement and these variables, as these variables are also expected to impact nurses’ work engagement, satisfaction, and productivity significantly.

## 5. Conclusions

Work engagement is the ability to enjoy and be dedicated to one’s work and can indicate the likelihood of an individual remaining in their current job. This study identified several personal and professional factors that affect nurses’ work engagement. These findings can be of great significance for researchers, healthcare leaders, and practitioners. Specifically, these findings can help organizations determine the factors that positively or negatively affect work engagement and allow them to focus on improving the enabling factors and suppressing the inhibitory ones. A replication of this study using a larger representative sample can ensure the generalizability and external validity of these findings.

## Figures and Tables

**Table 1 healthcare-11-00572-t001:** Participants’ Characteristics (*n* = 460).

Selected Factors		*n*	Percentage of Total Sample
Age (yrs.)	20 < x < 30	215	46.7
	30 < x < 40	198	43.1
	40 < x < 50	39	8.5
	50+	8	1.7
Gender	Female	402	97.4
	Male	58	2.6
Highest Educational Level	Diploma	34	7.4
	BSN *	413	89.8
	Higher education (master’s degree or PhD)	13	2.9
Participation in committees and work teams	Yes	276	60
	No	184	40
Nationality	Saudi	50	10.9
	Non-Saudi	410	89.1
Current work area	General ward	270	58.7
	Critical care unit	64	13.9
	Specialized area	126	27.5
Experience (yrs.)	1 < x < 5	80	17.4
	5 < x < 10	248	53.9
	10 < x < 20	110	23.9
	20+	22	4.8

* BSN: Bachelor of Science in Nursing.

**Table 2 healthcare-11-00572-t002:** Nurses’ Level of Work Engagement.

Work Engagement	Mean	SD *	Level
Dedication	5.68	1.027	High
Vigor	5.44	0.959	High
Absorption	5.27	0.968	High
Total	5.47	0.908	High

* SD: standard deviation.

**Table 3 healthcare-11-00572-t003:** Association between Level of Work Engagement and Personal Factors.

Personal Factors	Work Engagement			
	Vigor	Dedication	Absorption	Total
Gender	5.44 ± 0.96	5.68 ± 1.03	5.27 ± 0.97	5.47 ± 0.91
Male	5.49 ± 1.03	5.67 ± 1.09	5.14 ± 1.14	5.45 ± 0.99
Female	5.44 ± 0.95	5.68 ± 1.02	5.29 ± 0.94	5.48 ± 0.90
T-test (*p*-value)	0.370 (0.711)	−0.052 (0.959)	−1.082 (0.280)	−0.222 (0.824)
Age (yrs.)	5.44 ± 0.96	5.68 ± 1.03	5.27 ± 0.97	5.47 ± 0.91
20 < x < 30	5.27 ± 1.01	5.46 ± 1.11	5.11 ± 0.95	5.29 ± 0.95
30 < x < 40	5.53 ± 0.90	5.82 ± 0.92	5.35 ± 0.95	5.58 ± 0.85
40+	5.87 ± 0.80	6.07 ± 0.81	5.67 ± 0.99	5.88 ± 0.74
F (*p*-value)	9.548 (0.001)	10.644 (0.001 *)	7.918 (0.001 *)	11.128 (0.001 *)
Education	5.44 ± 0.96	5.68 ± 1.03	5.27 ± 0.97	5.47 ± 0.91
Diploma	5.44 ± 0.96	5.68 ± 1.01	5.25 ± 0.96	5.47 ± 0.90
BSN ^$^	5.49 ± 0.96	5.72 ± 1.15	5.52 ± 1.15	5.58 ± 1.04
Higher education	5.40 ± 1.07	5.33 ± 1.21	5.38 ± 0.74	5.37 ± 0.93
F (*p*-value)	0.058 (0.943)	0.765 (0.466)	1.332 (0.265)	0.320 (0.726)
Nationality	5.44 ± 0.96	5.68 ± 1.03	5.27 ± 0.97	5.47 ± 0.91
Saudi	5.26 ± 1.16	5.47 ± 1.25	5.17 ± 1.21	5.31 ± 1.16
Non-Saudi	5.46 ± 0.93	5.70 ± 0.99	5.28 ± 0.93	5.49 ± 0.87
T (*p*-value)	−1.397 (0.163)	−1.529 (0.127)	−0.789 (0.431)	−1.379 (0.168)

* Correlation is significant at the 0.05 level. ^$^ BSN: Bachelor of Science in Nursing.

**Table 4 healthcare-11-00572-t004:** Association between Level of Work Engagement and Professional Factors.

Professional Factors	Work Engagement			
	Vigor	Dedication	Absorption	Total
Work setting	5.44 ± 0.96	5.68 ± 1.03	5.27 ± 0.97	5.47 ± 0.91
General ward	5.51 ± 0.94	5.73 ± 0.99	5.35 ± 0.98	5.54 ± 0.90
ICU	5.40 ± 0.89	5.54 ± 0.96	5.08 ± 0.87	5.36 ± 0.82
Specialized unit	5.31 ± 1.02	5.63 ± 1.12	5.19 ± 0.98	5.39 ± 0.96
F (*p*-value)	2.099 (0.124)	0.969 (0.380)	2.732 (0.066)	1.888 (0.153)
Years of experience	5.44 ± 0.96	5.68 ± 1.03	5.27 ± 0.97	5.47 ± 0.91
1 < x < 10	5.34 ± 0.98	5.56 ± 1.07	5.15 ± 0.96	5.36 ± 0.93
10 < x < 20	5.77 ± 0.75	6.08 ± 0.67	5.65 ± 0.84	5.85 ± 0.66
20+	6.06 ± 0.75	6.23 ± 0.84	5.86 ±1.08	6.06 ± 0.73
F (*p*-value)	11.084 (0.001 *)	11.835 (0.001 *)	12.870 (0.001 *)	14.081 (0.001 *)
Participation in committees	5.44 ± 0.96	5.68 ± 1.03	5.27 ± 0.97	5.47 ± 0.91
Yes	5.58 ± 0.90	5.79 ± 0.95	5.45 ± 0.92	5.61 ± 0.85
No	5.24 ± 1.01	5.51 ± 1.11	5.00 ± 0.98	5.27 ± 0.95
T (*p*-value)	3.707 (0.001 *)	2.854 (0.005 *)	5.031 (0.001 *)	4.102 (0.001 *)

* Correlation is significant at the 0.05 level. ICU: intensive care unit.

## Data Availability

All data generated or analyzed during this study are available from the second author Manar Alanazi and will be provided on request due to privacy/ethical restrictions.

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
