# Peer review of "Personal and Job-Related Factors Influencing the Work Engagement of Hospital Nurses: A Cross-Sectional Study from Saudi Arabia"

_healthcare, 2023, doi:10.3390/healthcare11040572_

Round 1

Reviewer 1 Report

General considerations:

We believe the paper is within the broad scope of the journal, it brings empirical data that contributes to the discussion on the antecedents of work engagement.

The writing is adequate, the title, abstract, and main sections of the paper are consistent with data presented. However, one of the keywords “turnover intention” should be eliminated since the research brings no data evidence regarding turnover intentions.

The aims are clearly stated and the method appropriate. The statistical analysis conducted is acceptable. The conclusions are generally supported by data presented.

Nevertheless, we have some questions that we believe the authors should address, before this paper is published, identified below with reference to the paper sections.

Introduction:

The introduction contextualizes the problem, with a literature review on the antecedents of job engagement on nursing professionals, stating the research problems. The structure of this section is adequate. However, we would encourage the authors to state clearly what is the novelty of their work, how does this paper contribute to the literature in the field?

Materials and Methods:

The design seams adequate for this type of study, and we are informed about sampling, data collection and the instrument used.

- Page 4, lines 197-198, you only present the Alpha for the total scale, the Alphas for the subscales should also be presented, I suggest including these values on table 2.

Results:

- Page 5, line 205-216 the actual sample and its description should be presented on the Methods section.

Discussion:

- Page 8, line 276, when it’s referred a “average level” you should state what is “average” in this particular context.

The limitations of the study are identified.

Conclusions:

The conclusions are in line with the results presented.

Author Response

Dear reviewer,

Thank you for your valuable revisions. We replied point by point to your suggestions. You can read our replies in the hereafter table.

#

Comment

Response

Page

Line

Reviewer: 1 

The writing is adequate, the title, abstract, and main sections of the paper are consistent with data presented. However, one of the keywords “turnover intention” should be eliminated since the research brings no data evidence regarding turnover intentions.

Thank you for your notice, the keyword eliminated

1

27

Introduction:

we would encourage the authors to state clearly what is the novelty of their work, how does this paper contribute to the literature in the field?

It's really a good point. I added the statements describing “how does this paper contribute to the literature in the field”

3

141-148

Materials and Methods:

- Page 4, lines 197-198, you only present the Alpha for the total scale, the Alphas for the subscales should also be presented, I suggest including these values on table 2.

the Alphas for the subscales added in the same part (Instrument). I couldn’t add them to table 2, since it may be confusing to the reader. I hope it is acceptable in it’s place

5

219-220

Results:

- Page 5, line 205-216 the actual sample and its description should be presented on the Methods section.

Thank you for your comment. The actual sample description and table 1 removed from “Results” section and added to the “Participants” section in the method

4-5

173-179

Discussion:

- Page 8, line 276, when it’s referred a “average level” you should state what is “average” in this particular context.

The statement revised and the word changed to be “a moderate level”, and a statement describing the cutoff values for the high-moderate-low levels of engagement added in the “instrument” section

8

5

306

206-209

Reviewer 2 Report

Comments

1. The empirical context of the analysis (i.e., focus on SA) should be motivated in the revised introduction.

2. The sample size that is used in the study is small (N=524). (See also point 8 below.)

3. The data section should be improved: 

a. Perceived working conditions might be related to non-response. This may cause bias to the estimates that are presented in the paper.

b. Do self-reported measures contain measurement error or not? Does this have implications for the interpretation of the estimation results?

4. The empirical analysis does not reveal causal effects.

5. The revised version should pay more attention to the quantitative size of the estimated effects.   

6. An important issue is that employees are not randomly assigned into workplaces, which may bias estimates for well-being at work (incl. employee engagement). This challenge can be addressed using information on employees’ wage and work histories (see https://doi.org/10.1016/j.jebo.2012.09.005). The issue should be stated in the revised version. 

7. The relationships can differ significantly e.g., by age. Small sample limits analysis (N=524).

8. What is the external validity of the estimation results for other contexts than SA?

Author Response

Dear reviewer,

Thank you for your valuable revisions. We replied point by point to your suggestions. You can read our replies in the hereafter table.

#

Comment

Response

Page

Line

Reviewer 2:

1. The empirical context of the analysis (i.e., focus on SA) should be motivated in the revised introduction.

The introduction section revised, and the empirical context (SA) motivated.

1

3

32

117-119

The sample size that is used in the study is small (N=524). (See also point 8 below.)

4

10

163-168

367-369

3. The data section should be improved:

a. Perceived working conditions might be related to non-response. This may cause bias to the estimates that are presented in the paper.

Thank you for the note.

This issue was included in the “Limitation” section

10

375-376

b. Do self-reported measures contain measurement error or not? Does this have implications for the interpretation of the estimation results?

Also, this point included in the limitation section

10

374-475

4. The empirical analysis does not reveal causal effects.

Thank you for the note, the nature of the relations is associative in nature, not causative relation, thus, I added a statement clarifying the further research work recommended.

10

386-387

5. The revised version should pay more attention to the quantitative size of the estimated effects. 

Revised and added to the limitation and further investigations

10

380-384

6. An important issue is that employees are not randomly assigned into workplaces, which may bias estimates for well-being at work (incl. employee engagement). This challenge can be addressed using information on employees’ wage and work histories (see https://doi.org/10.1016/j.jebo.2012.09.005). The issue should be stated in the revised version. 

This point also considered in interpreting the findings of the current study

8-9

296-300

7. The relationships can differ significantly e.g., by age. Small sample limits analysis (N=524).

Thank you for the significant note, and for that I added to the conclusion section a statement to replicate the study with larger samples.

10

397-398

8. What is the external validity of the estimation results for other contexts than SA?

Thank you for the question. Actually, since the sample was drown conveniently, a sampling bias is expected, and the findings cannot be generalized (external validity), thus, I added a statement recommending replication of the study on a larger and more representative sample.

10

397-398

Round 2

Reviewer 2 Report

I am happy with the paper.